# A comparison of non-magnetic and magnetic beads for measuring IgG antibodies against *Plasmodium vivax* antigens in a multiplexed bead-based assay using Luminex technology (Bio-Plex 200 or MAGPIX)

**Ramin Mazhari**[1,2], **Jessica Brewster**[1], **Rich Fong**[3], **Caitlin Bourke**[1,2], **Zoe S. J. Liu**[1,2], **Eizo Takashima**[4], **Takafumi Tsuboi**[4], **Wai-Hong Tham**[1,2], **Matthias Harbers**[5,6], **Chetan Chitnis**[7], **Julie Healer**[1,2], **Maria Ome-Kaius**[1,8], **Jetsumon Sattabongkot**[9], **James Kazura**[3], **Leanne J. Robinson**[1,2,8,10], **Christopher King**[3], **Ivo Mueller**[1,2,11]ᵒ, **Rhea J. Longley**[1,2]ᵒ*

1 Population Health and Immunity Division, Walter and Eliza Hall Institute of Medical Research, Parkville, Victoria, Australia, 2 Department of Medical Biology, University of Melbourne, Parkville, Victoria, Australia, 3 Case Western Reserve University, Cleveland, Ohio, United States of America, 4 Division of Malaria Research, Proteo-Science Center, Ehime University, Matsuyama, Japan, 5 CellFree Sciences Co., Ltd., Yokohama, Japan, 6 RIKEN Center for Integrative Medical Sciences, Yokohama, Japan, 7 Department of Parasites & Insect Vectors, Malaria Parasite Biology and Vaccines, Institut Pasteur, Paris, France, 8 Vector Borne Diseases Unit, PNG Institute of Medical Research, Madang, Papua New Guinea, 9 Mahidol Vivax Research Unit, Faculty of Tropical Medicine, Mahidol University, Bangkok, Thailand, 10 Burnet Institute, Melbourne, Australia, 11 Department of Parasites & Insect Vectors, Malaria Parasites & Hosts Unit, Institut Pasteur, Paris, France

ᵒ These authors contributed equally to this work.
* Longley.r@wehi.edu.au

## Abstract

Multiplexed bead-based assays that use Luminex® xMAP® technology have become popular for measuring antibodies against proteins of interest in many fields, including malaria and more recently SARS-CoV-2/COVID-19. There are currently two formats that are widely used: non-magnetic beads or magnetic beads. Data are lacking regarding the comparability of results obtained using these two types of beads, and for assays run on different instruments. Whilst non-magnetic beads can only be run on flow-based instruments (such as the Luminex® 100/200™ or Bio-Plex® 200), magnetic beads can be run on both these and the newer MAGPIX® instruments. In this study we utilized a panel of purified recombinant *Plasmodium vivax* proteins and samples from malaria-endemic areas to measure *P. vivax*-specific IgG responses using different combinations of beads and instruments. We directly compared: i) non-magnetic versus magnetic beads run on a Bio-Plex® 200, ii) magnetic beads run on the Bio-Plex® 200 versus MAGPIX® and iii) non-magnetic beads run on a Bio-Plex® 200 versus magnetic beads run on the MAGPIX®. We also performed an external comparison of our optimized assay. We observed that IgG antibody responses, measured against our panel of *P. vivax* proteins, were moderately-strongly correlated in all three of our comparisons (pearson r>0.5 for 18/19 proteins), however higher amounts of protein were required for coupling to magnetic beads. Our external comparison indicated that results

**Data Availability Statement:** All relevant data are within the paper and its Supporting Information files.

**Funding:** This work was supported by the National Health and Medical Research Council Australia (https://www.nhmrc.gov.au/) (#1092789, #1134989 and #1043345 to IM and #1143187 to W-HT), the National Institute of Allergy and Infectious Diseases (https://www.niaid.nih.gov/grants-contracts/opportunities) (NIH grant 5R01 AI 104822 to JS) and the Global Health Innovative Technology Fund (https://www.ghitfund.org/) (T2015-142 to IM). Additional funding directly supporting field studies was from the TransEPI consortium (supported by the Bill and Melinda Gates Foundation, https://www.gatesfoundation.org/). RJL is currently supported by an NHMRC Early Career Investigator Fellowship (1173210). W. H.T. is a Howard Hughes Medical Institute-Wellcome Trust International Research Scholar (https://www.hhmi.org/programs/biomedical-research/international-programs, 208693/Z/17/Z). We also acknowledge support from the National Research Council of Thailand. This work was made possible through Victorian State Government Operational Infrastructure Support and Australian Government NHMRC IRIISS. These funders had no role in study design, data collection and analysis, decision to publish, or preparation of the manuscript. Additional support for this study was provided by CellFree Sciences in the form of salary for MH. The specific roles of these authors are articulated in the 'author contributions' section.

**Competing interests:** I have read the journal's policy and the authors of this manuscript have the following competing interests: RJL, TT and IM are inventors on patent application PCT/US17/67926 on a system, method, apparatus and diagnostic test for Plasmodium vivax. MH is a paid employee of CellFree Sciences Co., Ltd. This does not alter our adherence to PLOS ONE policies on sharing data and materials.

generated in different laboratories using the same coupled beads are also highly comparable (pearson r>0.7), particularly if a reference standard curve is used.

## Introduction

Over the past 5–10 years there has been a rapid advancement of Luminex® bead-based technologies to measure antibody responses to multiple proteins simultaneously. These assays have numerous advantages over traditional enzyme-linked immuosorbent assays (ELISA), such as a reduction in sample volume required and reduced laboratory time if choosing multiple targets to assay for, as well as the main advantage of allowing multiplexed detection of antibody responses. This is particularly relevant for the detection of antibodies against complex pathogens that express many hundreds to thousands of proteins, such as the *Plasmodium* parasites (the causative agent of malaria). Access to standardized control reagents [1] will also allow results from these assays to be reliably compared between different laboratories, which may result in more consistent findings among different studies [2].

Multiplexed bead-based assays use Luminex® xMAP® technology [3] (https://www.luminexcorp.com/xmap-technology/), which centers on use of beads (microspheres) with different fluorescent colours that can be detected in unique regions on a compatible instrument such as a Luminex® 200™ (also known as a Bio-Plex® 200, sold by Bio-Rad), MAGPIX® or FLEXMAP 3D® (https://www.luminexcorp.com/xmap-instruments/). The beads are internally labeled with different ratios of two fluorophores, one in a red wavelength and the other infrared. The compatible instruments and related software have pre-gated channels that detect the internal fluorophores in discrete regions. Proteins of interest can be coupled to a unique set of beads, facilitating multiplexed detection of antibody responses to multiple proteins. Coupling is the process of attaching a specific protein to the bead, through carboxyl groups on the bead surface (covalent bonding). Several studies have been conducted with a focus on optimizing various steps of the coupling process or assay work-flow, in the context of detection of antibodies against *Plasmodium* proteins, such as bead coupling [4], sample pre-dilution [4], assay temperature [4], plate washing [4], operator expertise [4], incubation times [1], and bead numbers [5]. Two different types of bead compositions are available for coupling proteins: non-magnetic and magnetic. Non-magnetic beads can only be run on flow-based instruments such as the Luminex® 200™/Bio-Plex® 200 or FLEXMAP 3D®, whilst magnetic beads can be run on both flow-based instruments and the MAGPIX®. The MAGPIX® is based on CCD imaging technology, and offers some advantages over the flow-based systems such as reduced use of reagents such as sheath fluid and the reduced cost of the MAGPIX® instrument compared to the Luminex® 200™/Bio-Plex® 200 instruments.

The primary aim of this study was to perform a series of comparisons of both non-magnetic and magnetic beads and assaying those beads on the Bio-Plex® 200 or the MAGPIX®. A secondary aim was to determine whether this assay is reproducible in an independent laboratory through an external comparison. This study used a panel of 19 different *P. vivax* proteins and plasma samples from *P. vivax*-endemic areas to detect *P. vivax*-specific IgG responses.

## Materials and methods

### Plasma samples

For all assays described here, a pool of samples from individuals from Papua New Guinea (PNG) with high levels of anti-*Plasmodium* antibodies was used as a positive control for the

standard curve dilution to adjust for plate to plate variation, as previously described [6]. The standard curve was run on every plate.

Two sets of plasma samples from malaria-endemic areas were used for comparisons of non-magnetic and magnetic beads, and the different acquisition instruments. These were 80 individuals from a longitudinal observational cohort study in Thailand, conducted in the Kanchanaburi and Ratchaburi provinces in 2013–2014. This cohort has previously been described in detail [7, 8], and the 80 plasma samples used were collected at the last visit of the cohort. The second set of samples came from a longitudinal observational cohort study in the Solomon Islands, conducted on the island Ngella in 2013–2014. This cohort has previously been described in detail [7, 9], and 83 plasma samples were used from individuals at the last visit of this cohort.

An additional set of plasma samples from a cohort study in PNG was used for external comparison of the assay. Samples were selected from the Mugil II paediatric cohort study. The study enrolled 450 children aged 5–12 years old in 2012 from the Mugil area on the North Coast of Madang province. All children were given antimalarial drugs to eliminate blood-stage *Plasmodium* spp. and blood samples were collected for parasitological and immunological studies. For the external comparison, a set of 425 samples was used from the baseline time-point (collected 2 weeks after drug treatment).

### Ethics statement

All samples were collected after approval from local ethics committees, with volunteers/participants providing written informed consent and/or assent (and parents or guardians providing informed consent for children). The Ethics Committee of the Faculty of Tropical Medicine, Mahidol University, Thailand approved the Thai cohort study (MUTM 2013-027-01). The National Health Research and Ethics Committee of the Solomon Islands Ministry of Health and Medical Services (HRC12/022) approved the Solomon Islands study. The Mugil II paediatric cohort was approved by the PNG Institute of Medical Research Institutional Review Board (IMR IRB) (1116/1204), the PNG Medical Research Advisory Committee (MRAC) (11.21/1206), the Walter and Eliza Hall Institute Human Research Ethics Committee (WEHI HREC) (12/09), and the Case Western Reserve University Hospitals of Cleveland Medical Center (CWRU UHCMC) (05-11-11). The HREC at WEHI approved samples for use in Melbourne (#14/02).

### Coupling *P. vivax* proteins to non-magnetic and magnetic beads

The carboxylated beads were sourced from Bio-Rad (Bio-Plex® COOH Beads, 1ml, $1.25 \times 10^7$ beads/ml and Bio-Plex® Pro Magnetic COOH Beads, 1ml, $1.25 \times 10^7$ bead/ml) and stored at 2–4˚C. Optimisation of coupling procedures for non-magnetic and magnetic beads were done separately, due to the larger size of the magnetic beads generally requiring more protein (see Results). To be able to measure all plasma samples at the same dilution, we optimized all protein amounts by generating a log-linear standard curve with a positive control plasma pool from immune PNG donors (high responders to *Plasmodium* antigens). The positive control pool was used to generate a standard curve running from a 1/50 dilution to a 1/25,600 dilution (10 point standard curve, 2-fold serial dilution). One set amount of protein was selected that resulted in a log-linear standard curve over this dilution series; the amounts optimized are not saturating but enable one dilution of plasma (1/100) to be run for all samples. As different amounts of protein are coupled for each protein construct, the MFI cannot be directly compared between proteins. A representative standard curve for both non-magnetic and magnetic beads is shown in S1 Fig.

Coupling of *P. vivax* proteins to non-magnetic beads was performed as previously described [7]. Briefly, the optimised antigen concentration (Table 1) was coupled to $2.5 \times 10^6$ pre-activated

microspheres, in 100 mM monobasic sodium phosphate buffer pH 6.0, using 50mg/ml sulfo-NHS and 50 mg/ml of EDC to cross-link the proteins to the beads. The activated beads were washed and stored in PBS, 0.1% BSA, 0.02% Tween-20, 0.05% Na-azide, pH 7.4 at 4°C until use. For the coupling to magnetic beads, a magnet rack was used for pelleting the beads, instead of the centrifugation step for non-magnetic beads. The coupling is random, not directional, which is optimal when the epitopes within the proteins are unknown. We qualitatively assessed the stability of the coupled beads by visual comparison of the MFI of the standard curve over a nine-month period. A reduction in the MFI or loss of log-linearity were considered markers of instability.

*Plasmodium vivax* recombinant antigens were expressed and purified in three countries: Japan (Takafumi Tsuboi, Ehime University & Matthias Harbers, CellFree Sciences), Australia (Wai-Hong Tham and Julie Healer, Walter & Eliza Hall Institute of Medical Research) and France (Chetan Chitnis, Institut Pasteur). Proteins were expressed either in the wheat-germ cell-free expression system (WGCF) or *E. coli*. See Table 1 for a complete list of proteins and the optimised amount coupled to non-magnetic and magnetic beads.

### Multiplexed assay for measurement of *P. vivax*-specific antibody responses

To measure the IgG levels, a multiplexed bead based assay was used, as previously described [7]. Briefly, antigen-specific IgG was detected by incubating 500 beads of each antigen per well with plasma diluted at 1:100, in a final volume of 100μl. Non-magnetic beads were washed using a vacuum manifold, whereas magnetic beads were washed using a magnetic plate washer. After the washings, 100μl of a 1:100 dilution of 0.5mg/ml PE-conjugated Donkey F

**Table 1. *P. vivax* proteins used in the comparison experiments, with the concentration of protein coupled per non-magnetic and magnetic beads indicated (note amount is listed per 1x10^6 beads, 2.5x10^6 beads were used for a bulk coupling).**

| Gene Annotation | Protein ID | Expression System | Protein Concentration (µg/ul) | Construct, amino acids (size) | Protein amount (µg/1x10^6) non-magnetic beads | Protein amount (µg/1x10^6) magnetic beads |
|---|---|---|---|---|---|---|
| RBP2b (P25) | PVX_094255 | *E. coli* | 4.15 | 161–1454 (1294) | 0.21 | 0.24 |
| MSP1-19 | PVX_099980 | WGCF | 1.55 | 1622–1729 (108) | 0.30 | 1.60 |
| RBP2b | PVX_094255 | WGCF | 2.06 | 1986–2653 (667) | 0.28 | 3.20 |
| RAMA | PVX_087885 | WGCF | 0.78 | 462–730 (269) | 0.06 | 0.48 |
| PvEBPII | KMZ83376.1 | *E. coli* | 10 | 109–432 (324) | 0.08 | 0.20 |
| SSA-s16 | PVX_000930 | WGCF | 0.41 | 31-end (110) | 0.40 | 0.80 |
| PvRIPR | PVX_095055 | *E. coli* | 1 | 552–1075 (524) | 0.40 | 0.80 |
| MSP3.10 | PVX_097720 | WGCF | 0.64 | 25-end (828) | 0.40 | 0.80 |
| Hyp. Protein | PVX_097715 | WGCF | 0.7 | 20-end (431) | 0.14 | 1.20 |
| PvDBPII (AH) | AAY34130.1 | *E. coli* | 0.6 | 1–237 (237) | 0.43 | 0.56 |
| MSP8 | PVX_097625 | WGCF | 0.39 | 24–463 (440) | 0.28 | 0.56 |
| Unspecified/ Pv-fam-a | PVX_112670 | WGCF | 1.13 | 34-end (302) | 0.45 | 0.90 |
| Pv-fam-a | PVX_096995 | WGCF | 1.7 | 61-end (420) | 0.34 | 1.20 |
| MSP3.3 | PVX_097680 | WGCF | 0.55 | 21-end (996) | 0.48 | 0.32 |
| MSP7.1 | PVX_082700 | WGCF | 0.33 | 23-end (397) | 0.40 | 0.60 |
| MSP5 | PVX_003770 | WGCF | 0.58 | 23–365 (343) | 0.01 | 0.016 |
| MSP7 | PVX_082670 | WGCF | 0.61 | 24-end (388) | 0.40 | 0.40 |
| PvTRAP/ SSP2 | PVX_082735 | WGCF | 0.9 | 26–493 (468) | 0.40 | 0.80 |
| PvDBPII (sal1) | PVX_110810 | *E. coli* | 1.2 | 193–521 (329) | 0.29 | 0.24 |

Gene annotations and protein IDs were sourced from PlasmoDB (release 36, http://plasmodb.org/plasmo/), or GenBank when necessary. All proteins have previously been used and described in our past work [7].

(ab)2 anti-human IgG (JIR 709-116-098) was added. At least 15 beads of each region/antigen were then acquired and analysed on a Bio-Plex® 200 instrument and/or a MAGPIX® instrument as per the manufacturer's instructions. Note that for comparing data between Bio-Plex® 200 and MAGPIX® instruments it is important that the "high RP1" target is not selected on the Bio-Plex® 200, as this option is not available on the MAGPIX®. On each plate, a twofold serial dilution from 1/50 to 1/25,600 of a seropositive control plasma pool (generated from PNG adults) was included. Note that for the external comparison both labs used the same PNG control pool to generate the standard curve. Each instrument was maintained as instructed by the manufacturer, with the relevant calibration, validation and/or verification beads run daily or as indicated by the manufacturer. Note that the Bio-Plex® calibration beads used were different between the two laboratories for the external comparison.

The results were expressed as mean fluorescence intensity (MFI) of at least 15 beads for each antigen. We have previously determined that data from at least 15 beads is required per antigen for consistent and repeatable results.

## Instruments

Antibody measurements were acquired using a Bio-Plex® 200 Multiplexing Analyzer System from Bio-Rad for all non-magnetic coupled beads (Bio-Plex® 200System, Bio-Plex® high-throughput fluidics system, microplate platform and a computer with the Bio-Plex® manager software v.5.0). Washing steps were carried out on a Bio-Rad Aurum vacuum manifold.

For all magnetic coupled beads a MAGPIX® Multiplexing System from Millipore was used (MAGPIX® System and the Xponent software V.4.2). Washing steps were carried out using a magnetic plate washer from BioTek Instruments (BioTek ELx50). A Bio-Rad Sure Beads magnetic rack was used during the coupling process.

Plates were incubated on a Ratek Platform shaker (Microtiter/PCR Plate Shaker). A Vortex Sonicator (Branson 2200), a BioSan Vortex V-1 plus and a Table centrifuge (Eppendorf Centrifuge 5424) were also used during the coupling process.

## Statistical analysis

The raw MFI results were converted to relative antibody units (RAU) using protein-specific standard curve data (see S1 Fig for examples of standard curves). A log–log model was used to obtain a more linear relationship, and a five-parameter logistic function was used to obtain an equivalent dilution value compared to the PNG control plasma. We extrapolated one step further beyond the lowest dilution (i.e. from 1/25,600 to 1/51,200), resulting in converted data units ranging from $1.95 \times 10^{-5}$ to 0.02, as previously described [7]. This was performed in R. Pearson's r correlations were performed to determine the strength of correlation and the statistical significance for all comparisons. To enable these parametric correlations, data were log-transformed prior to the analysis to better fit the normal distribution. Pearson r values <0.3 were considered weak, 0.3–0.7 moderate, and >0.7 strong correlations.

## Results and discussion

### Comparison of total IgG antibodies detected against *P. vivax* antigens coupled to either non-magnetic or magnetic beads and assayed by a Bio-Plex® 200 instrument

Total IgG antibody levels against a panel of 19 *P. vivax* proteins, measured in plasma samples from 163 individuals living in malaria-endemic areas of Thailand and the Solomon Islands, were assayed using either non-magnetic or magnetic beads and run on a Bio-Plex® 200

instrument. IgG levels to 18 of 19 proteins were moderate-strongly correlated between non-magnetic and magnetic assays, with Pearson r-values ranging from 0.53–0.98 (all p<0.0001) (Fig 1), supporting previous findings based on *P. falciparum* proteins [10]. This is despite different amounts of each protein being coupled to non-magnetic versus magnetic beads (Table 1). The exception was for the protein PVX_003770 (MSP5), with the lowest correlation coefficient at r = 0.27 (p<0.001). A sub-set of the samples that had relatively high antibody levels for PVX_003770 when assayed with non-magnetic beads had relatively low antibody levels when assayed with magnetic beads, likely accounting for the low correlation coefficient observed. Interestingly, the amount of protein coupled for PVX_003770 (for both non-magnetic and magnetic beads) was substantially lower than for the other proteins. Future experiments are planned to determine whether increasing the protein amount for PVX_003770 could result in a higher correlation between the two platforms.

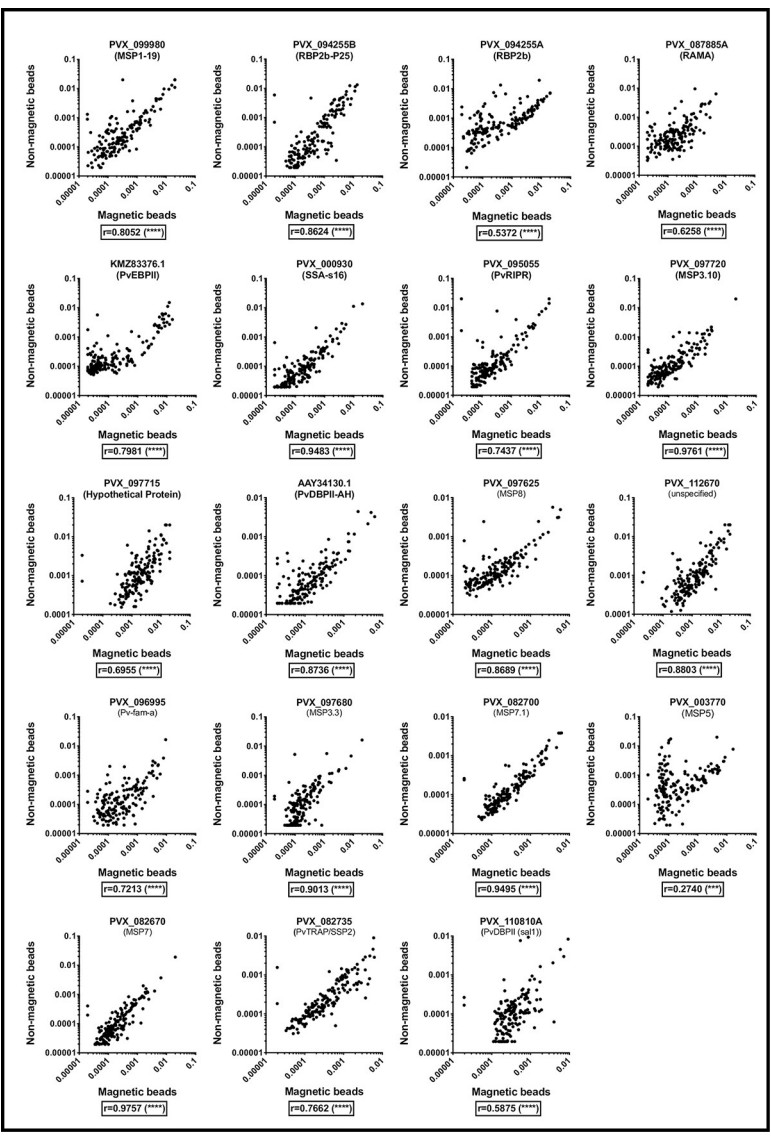

**Fig 1. IgG antibody levels (RAU) measured against 19 *P. vivax* proteins in samples from malaria-endemic areas, using either non-magnetic or magnetic beads and run on a Bio-Plex® 200 instrument.** *** p<0.001, **** p<0.0001.

## Comparison of total IgG antibodies detected against *P. vivax* antigens coupled to magnetic beads and assayed using either a Bio-Plex® 200 instrument or a MAGPIX® instrument

For this comparison, all 19 *P. vivax* antigens were coupled to magnetic beads only, at the optimised antigen concentrations. Total IgG antibody levels were measured in the same set of 163 plasma samples, with the assay run on both a Bio-Plex® 200 and a MAGPIX® instrument. To our knowledge, this is the first published report of this comparison. Here, the Pearson r correlation coefficients indicated a strong correlation between samples run on both instruments (r = 0.985–0.999, p<0.0001, Fig 2). These results indicate that results obtained on either platform, when antigens are coupled at the same optimised concentrations to magnetic beads, are highly comparable. The strength of the correlations in this comparison is stronger than the previous analysis (which compared non-magnetic versus magnetic beads on the same instrument), presumably because the same sets of coupled beads were run on both instruments. The

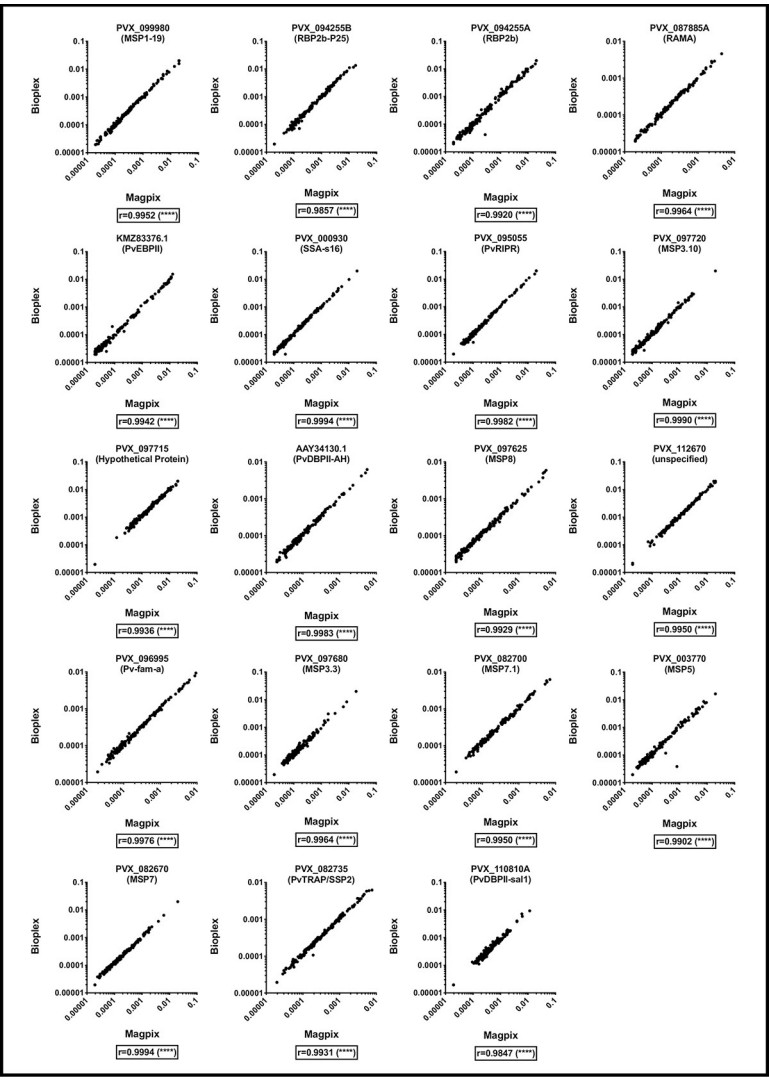

**Fig 2. IgG antibody levels (RAU) measured against 19 *P. vivax* proteins in samples from malaria-endemic areas, using magnetic beads and run on either a Bio-Plex® 200 instrument or MAGPIX® instrument.** **** p<0.0001.

strength of the correlations suggests that results obtained on the Bio-Plex® 200 and MAG-PIX® are interchangeable.

## Comparison of total IgG antibodies against *P. vivax* antigens coupled to non-magnetic beads and analyzed on a Bio-Plex® 200 instrument and antigens coupled to magnetic beads and analyzed on a MAGPIX® instrument

The final comparison we wanted to conduct was of antigens coupled to non-magnetic beads and assayed on a Bio-Plex® 200 instrument with antigens coupled to magnetic beads and assayed on a MAGPIX® instrument. As non-magnetic beads are cheaper to purchase, users that have only a Bio-Plex® 200 instrument would potentially favour this configuration (even though the instrument can run both non-magnetic and magnetic beads). Conversely, for users that only have a MAGPIX® instrument, they are only able to run magnetic beads as the instrument cannot detect non-magnetic beads. To our knowledge, this is the first published report of this comparison for a non-commercial assay.

It was again observed that there was a moderate-strong correlation between results obtained using the non-magnetic beads/Bio-Plex® 200 and magnetic beads/MAGPIX® platforms, with Pearson r correlation coefficients ranging from 0.42–98 ($p<0.0001$, Fig 3). These correlation coefficients are similar to those obtained in the first comparison (non-magnetic versus magnetic beads both run on the Bio-Plex® 200 instrument), and provide further support for our finding that antigens coupled to either type of beads and run on either instrument generally give very comparable total IgG measurements. As we observed in the first comparison, the weakest correlation was again for the protein PVX_003770 (with a moderate r correlation coefficient of 0.42).

## External comparison of a multiplexed assay using *P. vivax* antigens coupled to non-magnetic beads and analyzed on a Bio-Plex® 200 instrument

The results thus far indicate that IgG levels measured using either non-magnetic or magnetic beads and assayed on either a Bio-Plex® 200 or MAGPIX® instrument are highly comparable. A group of 3 staff members, but all at the same Institute (Walter & Eliza Hall Institute, WEHI) using the same instruments, performed these measurements. Therefore an additional comparison was performed: external comparison of the assay at an independent research Institute located overseas (Case Western Reserve University, CWRU). It is important to note that each Institute used their own (commercial) Bio-Plex® calibration and validation beads to set-up and maintain their respective instruments, which may contribute to some variation in results between laboratories.

A set of 425 plasma samples were aliquoted at CWRU and shared with WEHI. At the same time, a set of 12 *P. vivax* proteins (Table 2) were coupled to non-magnetic beads at WEHI and shared with CWRU. During the same week assays were performed to measure total IgG antibodies against these *P. vivax* antigens in the 425 plasma samples on Bio-Plex® 200 instruments independently at each Institute (total of 6 plates run at each Institute). After exclusion of plates or samples following quality control checks (positive control–non log-linear standard curve; bead counts < 15), data from 318 samples was directly compared between sites. The drop from 425 to 318 samples was largely due to one plate with failed standard curves that could not be repeated due to sample availability. IgG levels were compared first using raw data (MFI values). The Pearson r correlation coefficients indicated a strong correlation for all proteins with r-values ≥ 0.76 ($p<0.0001$), with the exception of PVX_094255 (RBP2b) (r = 0.57, $p<0.0001$) (Table 3, scatter plots in S2 Fig). The same correlation analysis was then performed on data

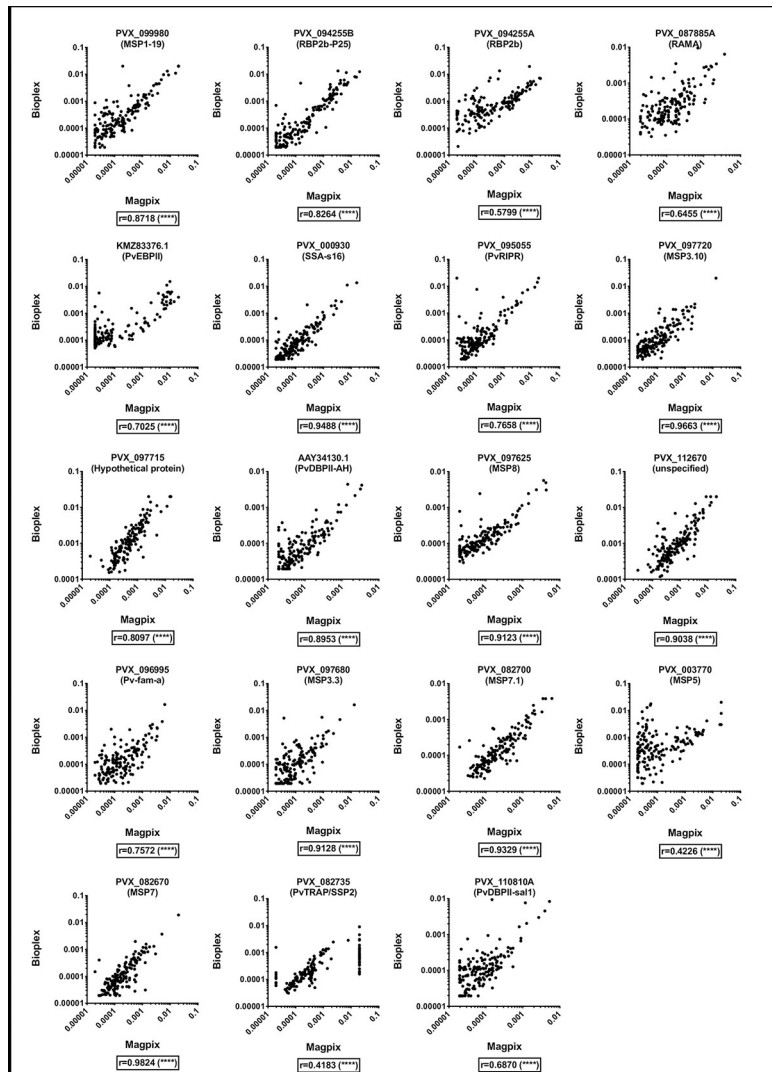

**Fig 3. IgG antibody levels (RAU) measured against 19 *P. vivax* proteins in samples from malaria-endemic areas, using non-magnetic beads and run on a Bio-Plex® 200 compared to use of magnetic beads run on a MAGPIX® instrument.** **** p<0.0001.

converted in R using the standard curves (to account for any plate-plate variation). Strong correlation coefficients were observed for all 12 proteins, including PVX_094255 (r values ≥ 0.72, p<0.0001) (Table 3, scatter plots in S3 Fig). For the majority of proteins, the correlation was stronger after conversion (Table 3). This is expected given the conversion, based on the standard curve generated with a plasma pool from immune PNG donors that is run on every plate, is used to account for any plate-plate variation and potentially to overcome differences that might be attributed to different machines maintained and set-up with different calibration and validation bead sets.

These results indicate that data generated using this multiplexed assay are highly reproducible in a different laboratory setting when the same coupled-beads are used, particularly if both laboratories have access to the same positive control for standardization. Unfortunately, whilst there is a WHO reference reagent for *P. falciparum* serology studies [11], there is not yet a

**Table 2. *P. vivax* proteins used for the external comparison.**

| Gene Annotation | Protein ID | Expression System |
|---|---|---|
| MSP1-19 | PVX_099980 | WGCF |
| Pv-fam-a | PVX_096995 | WGCF |
| hypothetical protein, conserved | PVX_094830 | WGCF |
| Pv-fam-a | PVX_112670 | WGCF |
| MSP7 | PVX_082650 | WGCF |
| RBP2b | PVX_094255 | WGCF |
| hypothetical protein, conserved | PVX_001000 | WGCF |
| merozoite surface protein 8 | PVX_097625 | WGCF |
| PvTRAP/SSP2 | PVX_082735 | WGCF |
| MSP7 | PVX_082645 | WGCF |
| PvRBP-2, putative | PVX_090330 | WGCF |
| sexual stage antigen s16 | PVX_000930 | WGCF |

Proteins were coupled to non-magnetic beads at WEHI and half of each batch of bead-conjugated protein was shipped to CWRU. All proteins have previously been used and described in our past work [7].

**Table 3. External comparison of the non-magnetic bead assay run on the Bio-Plex® 200.**

| Protein ID | Correlation MFI (n = 318) | Correlation RAU (n = 318) |
|---|---|---|
| PVX_099980 | 0.87 **** | 0.92 **** |
| PVX_096995 | 0.83 **** | 0.87 **** |
| PVX_094830 | 0.76 **** | 0.74 **** |
| PVX_112670 | 0.79 **** | 0.81 **** |
| PVX_082650 | 0.85 **** | 0.84 **** |
| PVX_094255 | 0.57 **** | 0.72 **** |
| PVX_001000 | 0.83 **** | 0.83 **** |
| PVX_097625 | 0.86 **** | 0.85 **** |
| PVX_082735 | 0.90 **** | 0.92 **** |
| PVX_082645 | 0.89 **** | 0.87 **** |
| PVX_090330 | 0.84 **** | 0.81 **** |
| PVX_000930 | 0.89 **** | 0.91 **** |

Pearson r correlation coefficients are shown for both the raw data (MFI) and the standard curve converted data (RAU).

**** p<0.0001.

similar product available for *P. vivax*. Importantly, we also assessed the stability of the coupled beads by running the standard curve 10 times over a period of 9 months (intensely for 2 months) (S4 Fig). For most proteins the coupled beads were highly stable (11/16 tested over 9-months), with the MFI dropping for three proteins and increasing for two proteins. This is supported by previous research that has indicated the stability of protein-coupled beads [10], noting that the stability may vary by antigen [12].

## Conclusions

The aim of this study was to determine whether multiplexing assays performed using magnetic beads or non-magnetic beads are highly comparable, independent of the beads and platform used to analyze the assays. We compared here a total of 19 *P. vivax* proteins that were coupled

to both magnetic beads and non-magnetic beads. The protein concentration used for the couplings was individually determined by optimisation for each protein for the chosen bead type (Table 1). For this, a dilution series from the positive control plasma pool, prepared from immune PNG donors, was used to generate a log-linear standard curve for each protein. The non-magnetic beads are 5.5μm in size, whilst the magnetic beads are 6.5μm in size, likely accounting for the need to couple on average 0.3μg of protein to non-magnetic versus 0.8 μg of protein to magnetic beads (per 1x10$^6$ beads). One coupling reaction using these amounts of protein is enough to assay > 3000 samples in singlicate, thus the slightly higher amount of protein required for magnetic beads is unlikely to be a limitation to using this format. We did not assess the efficiency of antigen coupling, which could potentially be an important variable impacting the amount of protein required for coupling.

We have demonstrated that results are moderately-strongly correlated whether using proteins coupled to magnetic beads or non-magnetic beads and analysed using either a Bio-Plex® 200 (non-magnetic and magnetic beads) or MAGPIX® (magnetic beads only). Our external comparison has also demonstrated that results generated in different laboratories are strongly correlated, if a reference standard curve is included for standardization. Therefore researchers can, in principle, compare data generated with a different type of bead or assayed using a different instrument platform, if the amount of protein coupled is optimised for the correct type of bead. Overall, the choice of assay platform and instrument used is up to the user. However, we do suggest that selecting one bead composition for running experiments is preferred, given the variation in strength of correlation between proteins (Fig 1). Running magnetic beads on a Bio-Plex® 200 or a MAGPIX® generates data that is so highly correlated they could be considered interchangeable (Fig 2). An important consideration is that up to 100 different proteins can be assayed simultaneously using non-magnetic beads and a Bio-Plex® 200 instrument, whereas the maximum is 50 proteins using a MAGPIX®. If less than 50 proteins will be used, the MAGPIX® instrument is cheaper and enables washing steps to be conducted with magnets, which improves both bead retention [10, 13] and speed of the assay.

For future use and development of the assay, we recommended that a reference laboratory provide both protein-coupled beads and a positive control, along with a Standard Operating Procedure for the assay. All protein-coupled beads should be tested for stability and researchers provided with an expiry date for their use, in addition to checking the performance of the standard curve before each use. This should ensure repeatable and comparable measurements are generated between different research groups. A key focus of *P. vivax* serology efforts should be to develop a standard WHO reference reagent for *P. vivax* that is available to any research group worldwide.

Whilst these results were obtained in the context of *P. vivax*-specific IgG responses in individuals from malaria-endemic areas, the large panel of proteins used and consistent results obtained for all proteins suggest these results can be applied to guide studies in other fields. Luminex® xMAP® technology has been used to measure antibody responses against other infectious pathogens, such as HIV and influenza [14, 15], to a variety of vaccine antigens such as tetanus toxoid [16], and more recently to SARS-CoV-2 [17–19].

## Supporting information

**S1 Fig. Representative example of standard curves generated for each *P. vivax* protein on non-magnetic and magnetic beads.** MFI = median fluorescent intensity. S1 –S10 = standard 1 to standard 12 (2 fold serial dilution of positive plasma pool, starting at 1/50 dilution). The data are converted from MFI to relative antibody units (RAU) using a five-parameter logistic function to obtain an equivalent dilution value compared to the PNG control plasma. For

example, an MFI of similar value to that of the 1/50 dilution of the standard curve would result in an RAU of around 0.02 (or 1/50). The RAU values therefore range from $1.95 \times 10^{-5}$ (equivalent to 1/51,200 or S11, as the curve is extrapolated one step further) to 0.02.
(TIF)

**S2 Fig. Comparison of IgG antibody levels against 12 *P. vivax* proteins when run at WEHI compared to CWRU: Raw MFI values.**
(TIF)

**S3 Fig. Comparison of IgG antibody levels against 12 *P. vivax* proteins when run at WEHI compared to CWRU: Converted RAU values.**
(TIF)

**S4 Fig. Stability of protein-coupled magnetic beads over 9-months.** The original coupled beads were tested at every week for 2 months after coupling, then again at 9 months post-coupling. The MFI of the standard curves are presented (S1 = 1/50, then 2-fold serial dilution). New vials of secondary antibodies were opened on 19/02/19, 26/02/19 and 08/03/19. Protein PVX_094255 (WGCF construct) was not tested in this experiment.
(TIF)

## Acknowledgments

We wish to acknowledge the extensive field-teams in Thailand, Solomon Islands and PNG that originally collected samples in the studies that were used for this project. We thank Connie Li-Wai-Suen for providing the R code for the standard curve transformation.

## Author Contributions

**Conceptualization:** Leanne J. Robinson, Christopher King, Ivo Mueller, Rhea J. Longley.

**Data curation:** Ramin Mazhari, Jessica Brewster, Rich Fong, Rhea J. Longley.

**Formal analysis:** Ramin Mazhari, Jessica Brewster, Rhea J. Longley.

**Funding acquisition:** Takafumi Tsuboi, Matthias Harbers, Jetsumon Sattabongkot, Christopher King, Ivo Mueller, Rhea J. Longley.

**Investigation:** Ramin Mazhari, Jessica Brewster, Rich Fong, Caitlin Bourke, Zoe S. J. Liu, Maria Ome-Kaius, Jetsumon Sattabongkot, James Kazura, Leanne J. Robinson, Christopher King, Ivo Mueller, Rhea J. Longley.

**Methodology:** Ramin Mazhari, Jessica Brewster, Rich Fong, Caitlin Bourke, Zoe S. J. Liu, Christopher King, Ivo Mueller, Rhea J. Longley.

**Project administration:** Ivo Mueller, Rhea J. Longley.

**Resources:** Eizo Takashima, Takafumi Tsuboi, Wai-Hong Tham, Matthias Harbers, Chetan Chitnis, Julie Healer, Maria Ome-Kaius, Jetsumon Sattabongkot, James Kazura, Leanne J. Robinson, Ivo Mueller.

**Supervision:** Leanne J. Robinson, Christopher King, Ivo Mueller, Rhea J. Longley.

**Visualization:** Ramin Mazhari, Rhea J. Longley.

**Writing – original draft:** Ramin Mazhari, Ivo Mueller, Rhea J. Longley.

**Writing – review & editing:** Jessica Brewster, Rich Fong, Caitlin Bourke, Zoe S. J. Liu, Eizo Takashima, Takafumi Tsuboi, Wai-Hong Tham, Matthias Harbers, Chetan Chitnis, Julie

Healer, Maria Ome-Kaius, Jetsumon Sattabongkot, James Kazura, Leanne J. Robinson, Christopher King, Ivo Mueller, Rhea J. Longley.

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
