## [Decision Letter · Decision Letter 0]

25 Aug 2020

PONE-D-20-24214

A comparison of non-magnetic and magnetic beads for measuring IgG antibodies against P. vivax antigens in a multiplexed bead-based assay using Luminex® technology (Bio-Plex®200 or MAGPIX®)

PLOS ONE

Dear Dr. Longley,

Thank you for submitting your manuscript for review to PLoS ONE. After careful consideration, we feel that your manuscript will likely be suitable for publication if it is revised to address the points raised by the reviewers.  A significant number of topics need to be clarified and manuscript should be adjusted as suggested.   A Major concern was related to data analysis that should be revised as requested; perhaps controls using monoclonal antibodies could confirm the integrity of the exposed epitopes.  Finally, the authors should follow the policy of Plos One to share the raw data underlying their results.  Such policies help increase the reproducibility of the published literature, as well as make a larger body of data available for reuse and re-analysis. For your guidance, a copy of the reviewers' comments was included below. 

We look forward to receiving your revised manuscript.

Kind regards,

Luzia Helena Carvalho, Ph.D.

Academic Editor

PLOS ONE

Journal Requirements:

2. Thank you for disclosing that RJL, TT and IM are inventors on patent PCT/US17/67926 on a system, method, apparatus and diagnostic test for Plasmodium vivax in your competing interest section. As this is an international patent application and not a granted patent, please revise this statement to say that RJL, TT and IM are inventors on "patent application" PCT/US17/67926.

3. Please provide additional details regarding participant consent. In the ethics statement in the Methods and online submission information, please clarify whether you obtained informed consent from parents or guardians of participants in the pediatric cohort.

4. Thank you for stating the following in the Financial Disclosure section:

'This work was supported by the National Health and Medical Research Council Australia (https://www.nhmrc.gov.au/) (#1092789, #1134989 and #1043345 to IM and #1143187 to W-HT), the National Institute of Allergy and Infectious Diseases (https://www.niaid.nih.gov/grants-contracts/opportunities) (NIH grant 5R01 AI 104822 to JS) and the Global Health Innovative Technology Fund (https://www.ghitfund.org/) (T2015-142 to IM). Additional funding directly supporting field studies was from the TransEPI consortium (supported by the Bill and Melinda Gates Foundation, https://www.gatesfoundation.org/). RJL is currently supported by an NHMRC Early Career Investigator Fellowship (1173210). W.H.T. is a Howard Hughes Medical Institute-Wellcome Trust International Research Scholar (https://www.hhmi.org/programs/biomedical-research/international-programs, 208693/Z/17/Z).  We also acknowledge support from the National Research Council of Thailand. This work was made possible through Victorian State Government Operational Infrastructure Support and Australian Government NHMRC IRIISS. The funders had no role in study design, data collection and analysis, decision to publish, or preparation of the manuscript.'

We note that one or more of the authors are employed by a commercial company: CellFree Sciences Co., Ltd

c. We note that you have a patent relating to material pertinent to this article.

Please declare this patent (with details including name and number) in your amended Competing Interests statement, along with any other relevant declarations relating to employment, consultancy, patents, products in development or modified products etc.

Please confirm that this does not alter your adherence to all PLOS ONE policies on sharing data and materials, as detailed online in our guide for authors http://journals.plos.org/plosone/s/competing-interests by including the following statement: "This does not alter our adherence to  PLOS ONE policies on sharing data and materials.” If there are restrictions on sharing of data and/or materials, please state these. Please note that we cannot proceed with consideration of your article until this information has been declared.

Reviewers' comments:

Reviewer's Responses to Questions

**Comments to the Author**

1. Is the manuscript technically sound, and do the data support the conclusions?

Reviewer #1: Partly

Reviewer #2: Yes

Reviewer #3: Partly

2. Has the statistical analysis been performed appropriately and rigorously? 

Reviewer #1: No

Reviewer #2: Yes

Reviewer #3: Yes

3. Have the authors made all data underlying the findings in their manuscript fully available?

Reviewer #1: Yes

Reviewer #2: Yes

Reviewer #3: Yes

4. Is the manuscript presented in an intelligible fashion and written in standard English?

Reviewer #1: Yes

Reviewer #2: Yes

Reviewer #3: Yes

5. Review Comments to the Author

Reviewer #1: The authors describe a systematic comparison of IgG antibody data as collected against a panel of 19 P. vivax antigens and human plasma samples from three endemic settings. This is an important study with the increase in use of the bead-based multiplex platform as performed by multiple groups worldwide, and the need to assess comparability among bead and platform variations. The study does well to directly compare variables of interest, but statistical presentation of results is only limited to correlation coefficients and p values inappropriately presented. The comparison should also include slope estimates for the regression models and p values corresponding to those.

Major concerns:

In the Introduction, the authors could do a better job of explaining the xMAP technology to the lay person unfamiliar with bead-based multiplex serology. Some suggestions are below, but review and imagine you’ve never heard of the multiplexing technology before when revising.

Table 1 and Table 2. For the benefit of the reader, for antigens that had been published on before, please include appropriate reference(s).

Throughout the paper, the authors state what appear to be p values for the r^2 correlation coefficient, which is inappropriate since r simply indicates goodness of fit for the regression model. For each comparison in the study, the authors should report both the r^2 as well as the slope, and can report on the p value of the slope. Additionally, if the slope deviates from 1.0, this is important information for the reader for generalizing if one bead type or platform would give consistently higher/lower MFI values. Instead of presenting all this information in the Figure panels, perhaps Tables (in a similar manner to Table 3) would make it easiest to digest. The authors should also state in the Methods what is criteria for ‘strong’ correlation.

In Conclusions, the authors accentuate the good correlation of responses between the various two comparators, but do not address potential reasons for when correlation (and slope) is not good. For example, why is MSP5 appear to have more issues than other Pv antigens? Many of the scatterplots in Figs 1 and 3 have very poor correlation and some points with differences >2 orders of magnitude, and this should be addressed.

Minor concerns:

- In title, spell out entire parasite name

- Throughout the document, if choosing to use the (R) symbol for company information, use consistently each time company is mentioned

- Abstract: instead of just mentioning responses were ‘strongly correlated’ also provide quantitative measures of correlation/difference (r^2 and slope estimates)

- Line 51-52: “such as a reduction in sample volume required and reduced laboratory time” if choosing multiple targets to assay for

- Line 57: “in more consistent findings among different studies”

- Line 59-62: if wanting to discuss the xMAP technology, need to be specific here about the IR dyes and pre-gating bead regions to allow multiplex data collection. Also need to mention the FlexMAP platform here. Provide a link to manufacturer’s website that explains this technology so readers can investigate for themselves.

- Line 62: explain what ‘coupling’ is

- Line 67: “Two different types of bead compositions are available…”

- Line 69-72: need to mention FlexMAP 3D system here

- Line 70-71: “offers advantages over the flow-based systems such as faster acquisition time” not true, and this is dependent on how many beads present in each assay well. Flow-based machines can actually read a plate faster than a MAGPIX

- Line 75-77: “A secondary aim was to demonstrate that this assay is highly reproducible in an independent laboratory through an external validation.” As currently worded, this isn’t an aim, but a pre-determined conclusion.

- Line 78-80: “however the large number of proteins assessed and consistent results obtained, suggest these findings should be generalizable for optimization of the multiplexed bead-based assay for other pathogens.” Even though there’s 19 antigens on your panel, they’re only to one pathogen. Not appropriate to extrapolate to the other myriad of human pathogens from this data alone.

- Line 80-82: “This is important in the context of the ongoing SARS-CoV-2 pandemic, as multiple laboratory assays based on Luminex technology are under development [6-8].” I get it that it’s really cool to talk about nCoV right now, but this sentence has nothing to do with your Pv malaria study and should be removed.

- Line 133-134: need to state here what would indicate that the beads were not stable

- Line 153-155, 162: Luminex recommends at least 35 beads acquired per region. Need to state here why 15 was chosen for this study.

- Line 266: “bead counts < 15”

- For Table 3 data, this is more of an “external comparison” rather than validation work

- Table 4: the majority of these factors (cost, time, etc.) are relative for an institution and the SOP being used, and this table should be removed since many of the statements are subjective

Reviewer #2: R. Mazhari and colleagues report the results from a straight-forward study that compared results using the original non-magnetic bead Luminex (100/200 – BioPlex) method with the newer magnetic bead-MAGPIX approach, by measuring antibody (Ab) levels to a series of P. vivax antigens. Although many researchers have conducted a few comparative assays when switching from the non-magnetic to magnetic format, to this reviewer’s knowledge, this is the first study to conduct an in-depth comparison between the two methods. Overall, the study is well described, the results clearly presented, and the results confirm what most investigators have assumed, namely, the two methods give very similar (but not identical) results. The study will provide reassurance to the research community that the two methods provide similar answers.

The following comments are meant to be helpful for improving the manuscript.

1. Methods: Line 153. “a 1:100 dilution of PE-conjugated Donkey …. was added.” More information is needed: What was the concentration used? How many ul were added? (e.g., the reagent (1 mg/ml) was diluted 1:1000 and 50uL was added to each well.”

2. Line 123-111124: “…, we optimized all protein concentrations …… .” This reviewer is not exactly sure what the statement means. Does it mean you coupled various concentrations of each antigen to beads, created a log-linear curve, and then selected a concentration of antigen that would give you a specific MFI after coupling using a 1:100 dilution of the plasma standard? This point is important for understanding Table 1, as the amounts listed do not appear to be “saturating” concentrations. It would also be useful the authors commented if it is possible to compare MFI between antigens, i.e., if 10,000 MF1 for antigen X means there are more Ab to X than 5,000 MFI to Ab Y? Some researchers report that if the beads are coupled with equal amounts of antigen, then comparison can be made across antigens (although I’m not sure I agree), but the question could be addressed.

3. It is surprising that the authors did not mention calibration or verification beads (standard), which I believe differ between BioPlex and MAGIX. Some mention of this would be useful. Were the same calibration beads (standards) used in the external validation study? If so, one might have expected even better agreement in SFig. 1 for MFI. Some mention of the influence (or lack thereof) of instrument calibration should be included.

4. In sFig. 1, it is somewhat difficult to interpret/compare the results, especially between antigens, because different Y- and X- axes were used. For example, for MSP1 the Y-Axis is from 100 to 10,000, but the X-axis is from 10 to 10,000. If similar results were obtained between the two assays, one should be able to draw a diagonal (45o) line, and half the data points would be above and half below. So, having the same X- and Y- axes is beneficial. It appears that higher MFI were obtained at CASE for antigens 112670, MSP7, and MSP8 (majority data points above the diagonal line), but lower values for 096995, PVx-001000, 094830. Do the authors have an explanation for this?

5. Converting raw data to arbitrary units using a standard curve is always difficult for readers to truly envision. It would be beneficial if a standard curve was included in the Supplemental Information section and some discussion provided on how MFI were converted to relative antibody units (RAU). That is, in sFig. 1 the MFI values range from 10 to >30,000 MFI (i.e., numbers normally found in the literature; readers will be able to identify with the numbers). In SFig. 2, data were transformed using the standard curve and range from 0.00001 to 0.1 arbitrary units. How did you get these numbers? Based on the patterns in paired figures, it appears 10,000 MFI equals ~0.01 RAU, right? This area of the manuscript would benefit from clarification. Based on the data provided and the reproducibility of the results across the various formats, it seems that reporting data as MFI would be beneficial, since Luminex data produced in all laboratories around the world are generated as MFI. Since the data using the standard curve is reported to be only slightly better, the authors might consider including a discussion/comment about the process of data transformation using R compared to simply using MFI.

6. SFig. 1 Y-axis is labeled Case; whereas, in sFig. 2 it is labeled CWRU. The labels should have the same nomenclature.

Minor comments:

1. Abstract: Technically, the word data is a plural term (datum singular). So, Line 34 should read “Data are lacking …..”. However, the world may be changing in how this term is perceived.

2. Line 49: “…. Has been a rapid uptake of Luminex ……” The word “uptake” is colloquial, so suggest changing to another term, such as, increased, development, advancement, etc.

3. Line 54: Do Plasmodial parasites really express hundreds of thousands of proteins?

Just comments for the authors (FYI: no response needed)

1. Line 162: Our biostatisticians also calculated that 15 beads was the lowest number of beads needed to provide representative data. Alas, most investigators think that bead counts need to be ≥100. So, your manuscript will help set the record straight.

2. We have also found that antigen-coupled beads are quite stable. We have used the same batch of coupled beads, without loss of MFI, for over 10 years. We have also found that antigens tend to be more stable after they are coupled than if they are re-frozen and recoupled.

Reviewer #3: The authors presented a technical- descriptive paper which aimed to compared two different tools (beads) applied to serological multiplex methods. This bring interesting results that could help cutting steps on laboratory work and to choose the better technique for serological studies. The authors cited previews studies where beads-based multiplex assays have been used, suggesting that kind a technical is going to be the future for serological studies, as screening to evaluation of the antibody responses. But the standard immuno-assay (ELISA) is still a standard protocol to this kind of evaluation. Also, the use of a coupling beads needs more validations regarding the integrity of the protein conformation, once the coupled protein could hide some important epitope to antigen recognition by antibodies. Taken this and to make the present study afford to publication, I’d like to make some suggestions:

1. Show the standard curve for both type beads in the main manuscript. This is a big step to select the optimized antigen concentrations of the protein and beads necessary the assay. Also, will be interesting show the detail how MFI is converted to relative antibody units (RAU) using protein-specific standard curve data.

2. As the author are presenting a technical paper, showing a technique that could replace the ELISA, I suggest show the antibody titles do not change for coupled and non-coupled antigen for both type beads, using for example, the standard ELISA for at least an antigen with weak correlation (PVX_003770), and with an antigen with a high correlation, and decenter controls. Those technique aim to show the antibodies titles, which cannot be significant different compare with a standard immunological assay (ELISA).

3. I’m concern about the protein conformation preservation when it is coupled. Maybe a assay using monoclonal antibodies for which recognize the specific epitopes should be interesting to show that protein conformation are being preserved.

6. PLOS authors have the option to publish the peer review history of their article (what does this mean?). If published, this will include your full peer review and any attached files.

Reviewer #1: No

Reviewer #2: No

Reviewer #3: No

---

## [Author Response · Author response to Decision Letter 0]

15 Oct 2020

Reviewer #1: The authors describe a systematic comparison of IgG antibody data as collected against a panel of 19 P. vivax antigens and human plasma samples from three endemic settings. This is an important study with the increase in use of the bead-based multiplex platform as performed by multiple groups worldwide, and the need to assess comparability among bead and platform variations. The study does well to directly compare variables of interest, but statistical presentation of results is only limited to correlation coefficients and p values inappropriately presented. The comparison should also include slope estimates for the regression models and p values corresponding to those.

Major concerns:

In the Introduction, the authors could do a better job of explaining the xMAP technology to the lay person unfamiliar with bead-based multiplex serology. Some suggestions are below, but review and imagine you’ve never heard of the multiplexing technology before when revising.

We thank the reviewer for the suggestions below and have updated our introduction section on the xMAP technology.

Table 1 and Table 2. For the benefit of the reader, for antigens that had been published on before, please include appropriate reference(s).

As all proteins have been used and published by our group before, we have included this sentence in the Table headings: All proteins have previously been used and described in our past work [7].

Throughout the paper, the authors state what appear to be p values for the r^2 correlation coefficient, which is inappropriate since r simply indicates goodness of fit for the regression model. For each comparison in the study, the authors should report both the r^2 as well as the slope, and can report on the p value of the slope. Additionally, if the slope deviates from 1.0, this is important information for the reader for generalizing if one bead type or platform would give consistently higher/lower MFI values. Instead of presenting all this information in the Figure panels, perhaps Tables (in a similar manner to Table 3) would make it easiest to digest. The authors should also state in the Methods what is criteria for ‘strong’ correlation.

We thank the reviewer for this feedback. We think the confusion has come from our use of reporting r^2. This was an error and we should have reported the pearson r value (the r^2 indicates the goodness of fit of the model as the reviewer indicates). We ran a pearson parametric correlation on log-transformed data, and not a regression model, so will report the r. We have updated all Figures, table and text to include the pearson r value and not the r^2. We believe that the Figures are useful for the reader to interpret the data, and prefer to keep those rather than tables.

We have now added into the statistics section that we considered r values of <0.3 to be weak, 0.3 – 0.7 to be moderate, and > 0.7 to be indicative of a strong correlation.

In Conclusions, the authors accentuate the good correlation of responses between the various two comparators, but do not address potential reasons for when correlation (and slope) is not good. For example, why is MSP5 appear to have more issues than other Pv antigens? Many of the scatterplots in Figs 1 and 3 have very poor correlation and some points with differences >2 orders of magnitude, and this should be addressed.

We note that in the Results/Discussion we suggested that the potential reason for the lower correlation for MSP5 was the much smaller amount of antigen coupled to beads for this protein. Whilst this amount was optimised, perhaps the lower total amount of protein coupled leaves more space available on the beads for non-specific binding, resulting in differing results between the two bead compositions. We plan to investigate this further and potentially re-optimise coupling for this particular protein. We have changed our language in the discussion to more simply state that the correlations were moderate-strong, and suggest users stick to one type of bead to run their experiments if possible: “However, we do suggest that selecting one bead composition for running experiments is preferred, given the variation in strength of correlation between proteins (Fig 1). Running magnetic beads on a Bio-Plex® 200 or a MAGPIX® generates data that is so highly correlated they could be considered interchangeable (Fig 2).”

All minor concerns have been addressed as indicated below.

Minor concerns:

- In title, spell out entire parasite name

This has been done.

- Throughout the document, if choosing to use the (R) symbol for company information, use consistently each time company is mentioned

This has been done.

- Abstract: instead of just mentioning responses were ‘strongly correlated’ also provide quantitative measures of correlation/difference (r^2 and slope estimates)

This has been done.

- Line 51-52: “such as a reduction in sample volume required and reduced laboratory time” if choosing multiple targets to assay for

This has been added.

- Line 57: “in more consistent findings among different studies”

Between has been changed to “among”.

- Line 59-62: if wanting to discuss the xMAP technology, need to be specific here about the IR dyes and pre-gating bead regions to allow multiplex data collection. Also need to mention the FlexMAP platform here. Provide a link to manufacturer’s website that explains this technology so readers can investigate for themselves.

We have included the FLEXMAP 3D platform and added in a link to the Luminex Corp website, along with more information on the bead dyes and pre-gating on the software. 

- Line 62: explain what ‘coupling’ is

We have added: “Coupling is the process of attaching a specific protein to the bead, through carboxyl groups on the bead surface (covalent bonding).”

- Line 67: “Two different types of bead compositions are available…”

We have added the requested change to the sentence.

- Line 69-72: need to mention FlexMAP 3D system here

This has been added.

- Line 70-71: “offers advantages over the flow-based systems such as faster acquisition time” not true, and this is dependent on how many beads present in each assay well. Flow-based machines can actually read a plate faster than a MAGPIX

We thank the reviewer for this comment. Whilst our assay is run faster on a MAGPIX, it is true that this is highly dependent on the beads present in each assay well. We have removed this part of the sentence.

- Line 75-77: “A secondary aim was to demonstrate that this assay is highly reproducible in an independent laboratory through an external validation.” As currently worded, this isn’t an aim, but a pre-determined conclusion.

We have re-worded the aim.

- Line 78-80: “however the large number of proteins assessed and consistent results obtained, suggest these findings should be generalizable for optimization of the multiplexed bead-based assay for other pathogens.” Even though there’s 19 antigens on your panel, they’re only to one pathogen. Not appropriate to extrapolate to the other myriad of human pathogens from this data alone.

We appreciate the reviewers point and have removed this text.

- Line 80-82: “This is important in the context of the ongoing SARS-CoV-2 pandemic, as multiple laboratory assays based on Luminex technology are under development [6-8].” I get it that it’s really cool to talk about nCoV right now, but this sentence has nothing to do with your Pv malaria study and should be removed.

This has been removed.

- Line 133-134: need to state here what would indicate that the beads were not stable

This has been added: “A reduction in the MFI or loss of log-linearity were considered markers of instability.”

- Line 153-155, 162: Luminex recommends at least 35 beads acquired per region. Need to state here why 15 was chosen for this study. 

In our hands, with at least 15 beads our data is consistent and highly repeatable. We have added the sentence: “We have previously determined that data from at least 15 beads is required per antigen for consistent and repeatable results.”

- Line 266: “bead counts < 15”

Updated.

- For Table 3 data, this is more of an “external comparison” rather than validation work

We have changed reference to this work to “external comparison” throughout the paper. 

- Table 4: the majority of these factors (cost, time, etc.) are relative for an institution and the SOP being used, and this table should be removed since many of the statements are subjective

We appreciate this point and have removed the Table.

Reviewer #2: R. Mazhari and colleagues report the results from a straight-forward study that compared results using the original non-magnetic bead Luminex (100/200 – BioPlex) method with the newer magnetic bead-MAGPIX approach, by measuring antibody (Ab) levels to a series of P. vivax antigens. Although many researchers have conducted a few comparative assays when switching from the non-magnetic to magnetic format, to this reviewer’s knowledge, this is the first study to conduct an in-depth comparison between the two methods. Overall, the study is well described, the results clearly presented, and the results confirm what most investigators have assumed, namely, the two methods give very similar (but not identical) results. The study will provide reassurance to the research community that the two methods provide similar answers.

We thank the reviewer for the positive feedback on our paper and constructive comments below.

The following comments are meant to be helpful for improving the manuscript.

1. Methods: Line 153. “a 1:100 dilution of PE-conjugated Donkey …. was added.” More information is needed: What was the concentration used? How many ul were added? (e.g., the reagent (1 mg/ml) was diluted 1:1000 and 50uL was added to each well.”

This has been added.

2. Line 123-111124: “…, we optimized all protein concentrations …… .” This reviewer is not exactly sure what the statement means. Does it mean you coupled various concentrations of each antigen to beads, created a log-linear curve, and then selected a concentration of antigen that would give you a specific MFI after coupling using a 1:100 dilution of the plasma standard? This point is important for understanding Table 1, as the amounts listed do not appear to be “saturating” concentrations. It would also be useful the authors commented if it is possible to compare MFI between antigens, i.e., if 10,000 MF1 for antigen X means there are more Ab to X than 5,000 MFI to Ab Y? Some researchers report that if the beads are coupled with equal amounts of antigen, then comparison can be made across antigens (although I’m not sure I agree), but the question could be addressed.

We have added in further details as follows: “The positive control pool was used to generate a standard curve running from a 1/50 dilution to a 1/25,600 dilution (10 point standard curve, 2-fold serial dilution). One set amount of protein was selected that resulted in a log-linear standard curve over this dilution series; the amounts optimized are not saturating but enable one dilution of plasma (1/100) to be run for all samples.” In essence, we use a process of trial and error to determine an optimal amount of protein to couple to the beads, based on the standard curve generated from a 2-fold serial dilution of positive pool plasma. We also added the following statement: “As different amounts of protein are coupled for each protein construct, the MFI cannot be directly compared between proteins.”

3. It is surprising that the authors did not mention calibration or verification beads (standard), which I believe differ between BioPlex and MAGIX. Some mention of this would be useful. Were the same calibration beads (standards) used in the external validation study? If so, one might have expected even better agreement in SFig. 1 for MFI. Some mention of the influence (or lack thereof) of instrument calibration should be included.

We thank the reviewer for this comment; it was not something we had considered. We have added in more information in the methods on this point: “Each instrument was maintained as instructed by the manufacturer, with the relevant calibration, validation and/or verification beads run daily or as indicated by the manufacturer. Note that the Bio-Plex® calibration beads used were different between the two laboratories for the external comparison.” We have added this sentence in the results: “It is important to note that each Institute used their own (commercial) Bio-Plex® calibration and validation beads to set-up and maintain their respective instruments, which may contribute to some variation in results between laboratories.” We also updated the following sentence to include the part in italics: “This is expected given the conversion, based on the standard curve generated with a plasma pool from immune PNG donors, is used to account for any plate-plate variation and potentially to overcome differences that might be attributed to different machines maintained and set-up with different calibration and validation bead sets.”

4. In sFig. 1, it is somewhat difficult to interpret/compare the results, especially between antigens, because different Y- and X- axes were used. For example, for MSP1 the Y-Axis is from 100 to 10,000, but the X-axis is from 10 to 10,000. If similar results were obtained between the two assays, one should be able to draw a diagonal (45o) line, and half the data points would be above and half below. So, having the same X- and Y- axes is beneficial. It appears that higher MFI were obtained at CASE for antigens 112670, MSP7, and MSP8 (majority data points above the diagonal line), but lower values for 096995, PVx-001000, 094830. Do the authors have an explanation for this?

We appreciate the point made by the reviewer and have updated all Figures to ensure the Y and X-axes have the same scale. However, we believe that there should be some degree of variation that is random – so having some proteins have higher values and Case whilst some have higher values at WEHI is a good thing (if they were all higher at one Institute this would be a systematic bias). 

5. Converting raw data to arbitrary units using a standard curve is always difficult for readers to truly envision. It would be beneficial if a standard curve was included in the Supplemental Information section and some discussion provided on how MFI were converted to relative antibody units (RAU). That is, in sFig. 1 the MFI values range from 10 to >30,000 MFI (i.e., numbers normally found in the literature; readers will be able to identify with the numbers). In SFig. 2, data were transformed using the standard curve and range from 0.00001 to 0.1 arbitrary units. How did you get these numbers? Based on the patterns in paired figures, it appears 10,000 MFI equals ~0.01 RAU, right? This area of the manuscript would benefit from clarification. Based on the data provided and the reproducibility of the results across the various formats, it seems that reporting data as MFI would be beneficial, since Luminex data produced in all laboratories around the world are generated as MFI. Since the data using the standard curve is reported to be only slightly better, the authors might consider including a discussion/comment about the process of data transformation using R compared to simply using MFI.

We thank the reviewer for this comment and agree that some further detail would be beneficial in this section. We have updated this section to read: “The raw MFI results were converted to relative antibody units (RAU) using protein-specific standard curve data (see Figure S3 for examples of standard curves from non-magnetic bead couplings). A log–log model was used to obtain a more linear relationship, and a five-parameter logistic function was used to obtain an equivalent dilution value compared to the PNG control plasma. We extrapolated one step further beyond the lowest dilution (i.e. from 1/25,600 to 1/51,200), resulting in converted data units ranging from 1.95×10−5 to 0.02, as previously described [7]. This was performed in R.”

Whilst the correlations of data between WEHI and CWRU were quite good just when using the MFI, we believe that having a standard curve and converting the data to relative antibody units is beneficial for accounting for inter-operator variability and inter-assay variability.

6. SFig. 1 Y-axis is labeled Case; whereas, in sFig. 2 it is labeled CWRU. The labels should have the same nomenclature.

We have updated all to CWRU.

Minor comments:

1. Abstract: Technically, the word data is a plural term (datum singular). So, Line 34 should read “Data are lacking …..”. However, the world may be changing in how this term is perceived.

We have made this change.

2. Line 49: “…. Has been a rapid uptake of Luminex ……” The word “uptake” is colloquial, so suggest changing to another term, such as, increased, development, advancement, etc.

We changed to advancement.

3. Line 54: Do Plasmodial parasites really express hundreds of thousands of proteins?

It reads “many hundreds TO thousands” of proteins, which is true.

Reviewer #3: The authors presented a technical- descriptive paper which aimed to compared two different tools (beads) applied to serological multiplex methods. This bring interesting results that could help cutting steps on laboratory work and to choose the better technique for serological studies. The authors cited previews studies where beads-based multiplex assays have been used, suggesting that kind a technical is going to be the future for serological studies, as screening to evaluation of the antibody responses. But the standard immuno-assay (ELISA) is still a standard protocol to this kind of evaluation. Also, the use of a coupling beads needs more validations regarding the integrity of the protein conformation, once the coupled protein could hide some important epitope to antigen recognition by antibodies. Taken this and to make the present study afford to publication, I’d like to make some suggestions:

1. Show the standard curve for both type beads in the main manuscript. This is a big step to select the optimized antigen concentrations of the protein and beads necessary the assay. Also, will be interesting show the detail how MFI is converted to relative antibody units (RAU) using protein-specific standard curve data.

We have provided an additional supplementary figure (Figure S1 in the revised manuscript) that shows a representative example of the standard curve for each protein for each bead type. Following a similar comment from reviewer 2 we have provided more details of the standard curve conversion in the methods.

2. As the author are presenting a technical paper, showing a technique that could replace the ELISA, I suggest show the antibody titles do not change for coupled and non-coupled antigen for both type beads, using for example, the standard ELISA for at least an antigen with weak correlation (PVX_003770), and with an antigen with a high correlation, and decenter controls. Those technique aim to show the antibodies titles, which cannot be significant different compare with a standard immunological assay (ELISA).

We acknowledge the reviewers comment, however, we believe there is sufficient prior evidence that ELISA and Luminex assay results are correlated (i.e. https://jcm.asm.org/content/55/1/165#sec-2;
https://malariajournal.biomedcentral.com/articles/10.1186/s12936-018-2465-4#Sec16;
https://www.sciencedirect.com/science/article/pii/S0022175914000465?via=ihub), also noting that Luminex assays for some targets are more sensitive (https://malariajournal.biomedcentral.com/articles/10.1186/s12936-019-3027-0). Our aim for this manuscript was not to establish Luminex as a new assay, but rather to compare different versions of the assay that are available.

3. I’m concern about the protein conformation preservation when it is coupled. Maybe a assay using monoclonal antibodies for which recognize the specific epitopes should be interesting to show that protein conformation are being preserved.

Again we appreciate the reviewers comment but this is beyond the aims and scope of our manuscript. For most of the proteins we have tested, the epitopes are unknown (and mAbs have not been produced), so this would not be feasible to undertake. What we have done is to include further text that explains the binding to the beads is random, not directional, and that this is optimal when we don’t know the epitopes. “The coupling is random, not directional, which is optimal when the epitopes within the proteins are unknown.”

---

## [Decision Letter · Decision Letter 1]

12 Nov 2020

PONE-D-20-24214R1

A comparison of non-magnetic and magnetic beads for measuring IgG antibodies against Plasmodium vivax antigens in a multiplexed bead-based assay using Luminex® technology (Bio-Plex®200 or MAGPIX®)

PLOS ONE

Dear Dr. Longley,

Thank you for submitting your manuscript for review to PLoS ONE. After careful consideration, we feel that your manuscript will likely be suitable for publication if it is revised to address few points raised now by the reviewer. We therefore invite you to revise your manuscript paying close attention to the specific points detailed by the reviewers.  

We look forward to receiving your revised manuscript.

Kind regards,

Luzia Helena Carvalho, Ph.D.

Academic Editor

PLOS ONE

Reviewers' comments:

Reviewer's Responses to Questions

**Comments to the Author**

1. If the authors have adequately addressed your comments raised in a previous round of review and you feel that this manuscript is now acceptable for publication, you may indicate that here to bypass the “Comments to the Author” section, enter your conflict of interest statement in the “Confidential to Editor” section, and submit your "Accept" recommendation.

Reviewer #1: All comments have been addressed

Reviewer #2: (No Response)

Reviewer #3: All comments have been addressed

2. Is the manuscript technically sound, and do the data support the conclusions?

Reviewer #1: Yes

Reviewer #2: Yes

Reviewer #3: Yes

3. Has the statistical analysis been performed appropriately and rigorously? 

Reviewer #1: Yes

Reviewer #2: Yes

Reviewer #3: Yes

4. Have the authors made all data underlying the findings in their manuscript fully available?

Reviewer #1: Yes

Reviewer #2: Yes

Reviewer #3: Yes

5. Is the manuscript presented in an intelligible fashion and written in standard English?

Reviewer #1: Yes

Reviewer #2: Yes

Reviewer #3: Yes

6. Review Comments to the Author

Reviewer #1: (No Response)

Reviewer #2: The authors have made changes in the manuscript that enhance its clarity and addressed the issues raised by the reviewers. The study clearly makes the point that a correlation exists among the data generated using difference beads-based systems. The authors might consider the following suggestions:

1. The authors added the sentence: “This is expected given the conversion, based on the standard curve generated with a plasma pool from immune PNG donors…..” Was the standard curve run on each plate? If so, please add this information to the text.

2. Supporting Fig. 1 shows the linearity of the dilution curve. However, it is not totally clear how one obtains relative arbitrary unit (RAU) from the curve (MFI vs S1, S2, etc.). Could the authors please add information to the figure legend on how to convert the data from MFI to RAU?

3. Supportive Fig. 2 – YEAH! This figure shows the raw MFI and that subsequent statistical refinements not really needed. All readers will appreciate this figure.

4. Supportive Fig. 3 – shows the relationship of the adjusted values and documents the association after statistical adjustments. Interestingly, adjustments changed the correlations insignificantly. Nice to see the comparison.

5. Supplemental Figures 3 and 4 appear to be the same – comparison of RAU at CWRU and WHEI. Maybe the wrong figure was uploaded into the copy I received. Supportive Figure 4 doesn’t show changes in MFI over time. The authors need to check sFig. 4.

Reviewer #3: (No Response)

7. PLOS authors have the option to publish the peer review history of their article (what does this mean?). If published, this will include your full peer review and any attached files.

Reviewer #1: **Yes: **Eric Rogier

Reviewer #2: **Yes: **Diane Wallace Taylor

Reviewer #3: No

---

## [Author Response · Author response to Decision Letter 1]

15 Nov 2020

Reviewer #2: The authors have made changes in the manuscript that enhance its clarity and addressed the issues raised by the reviewers. The study clearly makes the point that a correlation exists among the data generated using difference beads-based systems. The authors might consider the following suggestions:

1. The authors added the sentence: “This is expected given the conversion, based on the standard curve generated with a plasma pool from immune PNG donors…..” Was the standard curve run on each plate? If so, please add this information to the text.

We have updated the sentence to read “This is expected given the conversion, based on the standard curve generated with a plasma pool from immune PNG donors that is run on every plate, is used to account for any plate-plate variation and potentially to overcome differences that might be attributed to different machines maintained and set-up with different calibration and validation bead sets.

We also updated the methods text to make it clear that the standard curve is run on every plate.

2. Supporting Fig. 1 shows the linearity of the dilution curve. However, it is not totally clear how one obtains relative arbitrary unit (RAU) from the curve (MFI vs S1, S2, etc.). Could the authors please add information to the figure legend on how to convert the data from MFI to RAU?

We have added the following text supporting Fig. 1 legend:

The data are converted from MFI to relative antibody units (RAU) using a five-parameter logistic function to obtain an equivalent dilution value compared to the PNG control plasma. For example, an MFI of similar value to that of the 1/50 dilution of the standard curve would result in an RAU of around 0.02 (or 1/50). The RAU values therefore range from 1.95×10−5 (equivalent to 1/51,200 or S11, as the curve is extrapolated one step further) to 0.02.

3. Supportive Fig. 2 – YEAH! This figure shows the raw MFI and that subsequent statistical refinements not really needed. All readers will appreciate this figure.

Thank you for your positive response!

4. Supportive Fig. 3 – shows the relationship of the adjusted values and documents the association after statistical adjustments. Interestingly, adjustments changed the correlations insignificantly. Nice to see the comparison.

Thank you for your positive response!

5. Supplemental Figures 3 and 4 appear to be the same – comparison of RAU at CWRU and WHEI. Maybe the wrong figure was uploaded into the copy I received. Supportive Figure 4 doesn’t show changes in MFI over time. The authors need to check sFig. 4.

Thank you for noticing this, we have replaced sFig. 4. with the correct figure.

---

## [Editor Report · Decision Letter 2]

18 Nov 2020

A comparison of non-magnetic and magnetic beads for measuring IgG antibodies against Plasmodium vivax antigens in a multiplexed bead-based assay using Luminex® technology (Bio-Plex®200 or MAGPIX®)

PONE-D-20-24214R2

Dear Dr. Longley,

We’re pleased to inform you that your manuscript has been judged scientifically suitable for publication and will be formally accepted for publication once it meets all outstanding technical requirements.

Kind regards,

Luzia Helena Carvalho, Ph.D.

Academic Editor

PLOS ONE
---

## [Editor Report · Acceptance letter]

23 Nov 2020

PONE-D-20-24214R2 

A comparison of non-magnetic and magnetic beads for measuring IgG antibodies against *Plasmodium vivax* antigens in a multiplexed bead-based assay using Luminex technology (Bio-Plex 200 or MAGPIX). 

Dear Dr. Longley:

I'm pleased to inform you that your manuscript has been deemed suitable for publication in PLOS ONE. Congratulations! Your manuscript is now with our production department. 

Kind regards, 

on behalf of

Dr. Luzia Helena Carvalho 

Academic Editor

PLOS ONE